# Heat Shock Protein Genes Affect the Rapid Cold Hardening Ability of Two Invasive Tephritids

**DOI:** 10.3390/insects15020090

**Published:** 2024-01-29

**Authors:** Yuning Wang, Yan Zhao, Junzheng Zhang, Zhihong Li

**Affiliations:** 1Department of Plant Biosecurity, College of Plant Protection, China Agricultural University, Beijing 100193, China; wynzb2333@163.com (Y.W.); zhaoyan1996@126.com (Y.Z.); zhangjz@cau.edu.cn (J.Z.); 2Key Laboratory of Surveillance and Management for Plant Quarantine Pests, Ministry of Agriculture and Rural Affairs, Beijing 100193, China

**Keywords:** heat shock protein genes, adaptation, invasive, tephritids, rapid cold hardening

## Abstract

**Simple Summary:**

*Bactrocera dorsalis* and *Bactrocera correcta* are two important fruit flies that are distributed in India and keep spreading to other countries due to global warming. To colonize other countries, they need to have the ability to adapt to winter temperatures, and it is thus important to know the factors behind their rapid cold hardening ability. In this study, we identified the temperature at which the two species showed the highest survival rate and performed transcriptome sequencing to explore the potential key genes involved in their rapid cold hardening ability. We identified four heat shock protein genes and tested their expression levels using RT-qPCR. Finally, we knocked down these genes to test their function in rapid cold hardening. We found that the survival rate after rapid cold hardening was related to the expression level of *Hsp68* and *Hsp70* in *B. dorsalis* and to that of *Hsp23* and *Hsp70* in *B. correcta*. This study helps elucidate the adaptation mechanism underlying the rapid cold acclimation-hardening ability of the two species and provides information for studies related to the mechanism of their tolerance to cold temperatures.

**Abstract:**

*Bactrocera dorsalis* and *Bactrocera correcta* are two invasive species that can cause major economic damage to orchards and the fruit import and export industries. Their distribution is advancing northward due to climate change, which is threatening greater impacts on fruit production. This study tested the rapid cold-hardening ability of the two species and identified the temperature associated with the highest survival rate. Transcriptome data and survival data from the two *Bactrocera* species’ larvae were obtained after rapid cold-hardening experiments. Based on the sequencing of transcripts, four Hsp genes were found to be affected: *Hsp68* and *Hsp70*, which play more important roles in the rapid cold hardening of *B. dorsalis*, and *Hsp23* and *Hsp70,* which play more important roles in the rapid cold hardening of *B. correcta*. This study explored the adaptability of the two species to cold, demonstrated the expression and function of four *Hsps* in response to rapid cold hardening, and explained the occurrence and expansion of these two species of tephritids, offering information for further studies.

## 1. Introduction

*Bactrocera dorsalis* and *Bactrocera correcta* are two invasive fruit fly species established in India [1,2], mainly through females laying eggs on host fruits, which affects fruit quality and yield and causes serious economic losses [3]. To date, control measures, such as sex lures and pesticides, have not completely removed or stopped their dispersal [4,5]. *B. dorsalis* is currently distributed across 75 countries in Asia, Africa, and Oceania, with a clear northward trend in recent years [6,7]. Currently, *B. correcta* is limited to several countries in North and Northeast Asia, although it is morphologically similar to and shares many hosts with *B. dorsalis* [8,9,10]. We suspect that this may be due to differences in their overwintering and colonizing abilities, which are related to the rapid cold-hardening mechanisms of the two *Bactrocera* species.

As ectothermic animals, the physiological, biochemical, and other activities of insects are influenced and limited by temperature. Temperature is also one of the most important factors influencing the geographical distribution and dispersal of insects [11]. *B. dorsalis* is expanding in its native and invasive range, in part because of global warming, which is evidence of their capacity to adapt to variable temperature environments. The ability of insects to increase their resistance to low-temperature damage is known as rapid cold hardening, which can increase the physiological resistance of insects to stressful conditions [12,13,14]. After rapid cold hardening, insects undergo a series of changes in their physiology, such as lowering their supercooling point and increasing the content of anti-freezing small molecules such as polyols in their bodies [15]. The adaptability of insects to resist cold may be linked to their ability to resist other adversities, and their capacity to inhibit growth in response to adverse temperature stimuli is essential for avoiding phenological mismatches and excessive metabolic demands [16,17]. A recent study showed that the gut microbiota promotes adult *B. dorsalis* resistance to cold stress by stimulating arginine and proline metabolism [18]. Histone modification, an epigenetic mechanism, may be involved in the thermal tolerance of *B. dorsalis*, but with different regulatory mechanisms in thermal hardening and more immediate temperature-dependent hardening responses [19]. Zhang provided insights into the evolutionary history of *B. dorsalis* through high-quality chromosome-level reference genome assembly and found that *Cyp6a9* was the most pleiotropic candidate gene involved in environmental adaptation [20]. However, the genes of the two *Bactrocera* species involved in rapid cold hardening are unknown.

Screening for differentially expressed genes using transcriptomics can help identify key genes in insects that respond to environmental stress. Variable expression of heat shock proteins (Hsps) is a regulatory mechanism that helps insects adapt to different environmental stresses by resisting environment-induced injuries [21]. In insects, members of the Hsp70 family have been implicated in the cold adaptability of *Drosophila melanogaster* [22]. In addition, Gu discovered that *Bactrocera* species have variable capacities to adapt to temperature and that different hardening responses may be related to the regulation of certain genes, such as *Hsp23*, which in turn may contribute to species distribution and invasive potential [23]. However, the role of heat shock proteins in the rapid cold hardening of *B. dorsalis* and *B. correcta* is not yet known.

Here, we tested the rapid cold hardening of larvae, selected the temperature with the highest survival rate for each of the two invasive species, *B. dorsalis* and *B. correcta*, and identified key genes associated with rapid cold hardening by analyzing transcriptome data. Finally, RNAi experiments were performed to determine the function of the genes identified. The objective of this work was to help elucidate the adaptation mechanism underlying the rapid cold hardening ability of the two species, resolve the confusion about differences in their geographic range, and provide information for further studies related to the mechanism of their tolerance to low temperatures.

## 2. Materials and Methods

### 2.1. Sample Collection

*B. dorsalis* was collected from Guangzhou (23.40° N, 113.22° E), Guangdong Province, China, and *B. correcta* was collected from Yuanjiang (23.60° N, 102° E), Yunnan Province, China. The two species were placed in an incubator with an artificial climate of 25 °C and 70% humidity and a 14L:10D light period, fed with water and artificial food (sugar/protein in a ratio of 3:1). The eggs were collected using a paper cup with holes and mango juice and rinsed with double-distilled water in a special larval medium. The medium contained sucrose, wheat bran, brewer’s yeast, sorbic acid, ascorbic acid, methyl p-aminobenzoate, and sterile water.

Two-milliliter centrifuge tubes were prepared with holes in their lids prior to treatment. The most temperature-sensitive developmental stage of the two species, the 3rd early instar larvae stage, was selected for the cold-hardening experiments. Thirty larvae and 4 g of solid food were placed in the tubes. Temperatures of 5–15 °C were chosen for rapid cold hardening based on the reference temperatures and supercooling points of the two species, and the contrast temperature was set at 25 °C [24]. Under 1 h of low temperature treatment, mortality of normal 3rd early instar larvae in the range 40–50% was selected as the extreme low temperature. The extreme low temperature of *B. dorsalis* was −7.8 °C, and that of *B. correcta* was −8.3 °C, based on our pre-experiment. The treatment method was 4 h of exposure at the hardening temperature and 1 h of recovery at 25 °C, then 1 h of exposure to the extreme low temperature and 4 h of recovery at 25 °C. Each group consisted of five replicates. After treatment, death or survival was determined by manipulation with forceps, and the survival rate was calculated. SPSS R25.0.0.0 was used for one-way analysis of variance and Tukey HSD tests.

### 2.2. RNA Extraction and Transcriptome Sequencing

For mRNA sequencing, *B. dorsalis* and *B. correcta* samples were collected from three groups of ten early third instar larvae after exposing them to hardening temperatures (5 °C and 9 °C) for 4 h, followed by a 1 h recovery period at 25 °C. Each group had four biological replicates. Total RNA was extracted using an miRNeasy Kit (Qiagen, Germantown, MD, USA) following the manufacturer’s protocol and sent to Lc-Bio to check RNA quality. After RNA extraction, the samples were sent to the LC-BIO company for further processing. Paired-end sequencing was performed on an Illumina Hiseq4000 LC Sciences platform, USA, following the vendor’s recommended protocol.

Sequence quality was verified using FastQC (http://www.bioinformatics.babraham.ac.uk/projects/fastqc/, accessed on 31 October 2020). All downstream analyses were based on clean, high-quality data. De novo transcriptome assembly was performed in Trinity 2.4.0 [25]. Transcripts were grouped into clusters based on their shared sequence content. All assembled unigenes were aligned against the non-redundant (Nr) protein database (http://www.ncbi.nlm.nih.gov/, accessed on 31 October 2020), Gene Ontology (GO) (http://www.geneontology.org, accessed on 31 October 2020), SwissProt (http://www.expasy.ch/sprot/, accessed on 31 October 2020), Kyoto Encyclopedia of Genes and Genomes (KEGG) (http://www.genome.jp/kegg/, accessed on 31 October 2020), and eggNOG (http://eggnogdb.embl.de/, accessed on 31 October 2020) databases. Databases using DIAMOND have an E-value threshold of <0.00001 [26]. For differentially expressed unigene analysis, salmon was used to determine the expression levels of unigenes by calculating TPM [27,28]. Differentially expressed unigenes were selected with log^2^ (fold change) >1 or log^2^ (fold change) <−1 and statistical significance (*p* < 0.05) using the R package edge R (3.17) [29].

### 2.3. Real-Time PCR and dsRNA Synthesis

To validate the results of the transcriptome analysis, we quantified the expression levels of the candidate genes *Hsp23*, *Hsp68*, *Hsp70*, and *Hsp27* in *B. dorsalis* and *B. correcta*. Thirty specimens from each group (two species at three temperature treatments, 25 °C, 5 °C, and 9 °C, for 4 h followed by 25 °C for a 1 h recovery, with five replicates per group) were randomly selected for analysis. RNA extraction was performed as described in Section 2.2. All cDNA was reverse transcribed using a PrimeScript RT Reagent Kit with a gRNA Eraser (Takara). Quantitative real-time PCR was performed on an ABI QuantStudio 6 Flex instrument (Applied Biosystems Europe, Brussels, Belgium) using SYBR Premix Ex Taq II (TliRNase H Plus; Takara, Japan). Moreover, *18s* was used as a control gene. Primer Premier 6.00 (Build 60006) was used to design the RT-qPCR primers (Appendix A). The reaction included 12.5 μL of SYBR Green mix, 1 μL of forward and reverse primers each, 0.5 μL of ROX Reference Dye II, 1 μL of cDNA, and 9 μL of ddH_2_O. The program was 95 °C for 30 s, followed by 40 cycles at 95 °C for 5 s and 60 °C for 34 s. Five biological replicates were carried out for statistical analysis. The 2-ΔCT method [30] and SPSS R25.0.0.0 were used to analyze the RT-qPCR data. The *t* test between each group was performed by SPSS as well. Figures were plotted using the OriginPro 2023b software (study version), and the group with the lowest gene expression was set as “1”.

Double-stranded RNA of HSP genes (*dsHSP*) was used to knock down the expression of target genes in *B. dorsalis* and *B. correcta*, with green fluorescent protein (*dsGFP*) used as a negative control. *dsRNA* was synthesized using the T7 RiboMAX Express RNAi system (Promega) with primers designated Hsps-dsRNA-F/R and Hsps-dsRNA-F/R-T7 (Appendix A). Larvae were fed 3 g of an artificial diet containing 30 µL of dsRNA solution (1000 ng/μL) in a 50-mL tube. Forty 3-day-old first-instar larvae for the cold response study and forty 4-day-old second-instar larvae for the rapid cold hardening study were collected and placed in 50-mL tubes that had three holes in the lid to allow air to pass through and that contained the artificial diet. Five replicates were performed for each treatment. The larvae were allowed to feed on the artificial diet containing *dsHsp* and *dsGFP* for 48 h and then transferred to tubes with a new artificial diet containing *dsRNA* for another 48 h. After 96 h of feeding, the larvae developed into early and late third-instar larvae.

### 2.4. Extreme Cold Stress and Hardening Response after dsHsp Feeding

After 96 h of *dsRNA* feeding, ten larvae each of *B. dorsalis* and *B. correcta* were sacrificed to perform *Hsp* gene expression analysis. The remaining 30 third-instar larvae were transferred to a 2-mL tube with food for low-temperature exposure. We chose early third-instar larvae to study the function of *Hsp* genes during rapid cold hardening because of their high survival rate at extreme temperatures [31]. The larvae were directly exposed to cold stress at −7.8 °C/−8.3 °C for 1 h, followed by 4 h at 25 °C. To determine the function of *Hsp* genes in rapid cold hardening, larvae were separately exposed to rapid cold hardening (5 °C and 9 °C) and control (25 °C) temperatures for 4 h and returned to 25 °C for 1 h. Ten of the larvae in each group were harvested for the study, while the remaining 30 larvae were exposed to cold stress at –7.8/–8.3 °C for 1 h, followed by a 25 °C recovery period for 4 h and an assessment of the survival rate. Each treatment was replicated five times. Data were analyzed using the *t* test in SPSS R25.0.0.0, with a comparison of *Hsps*’ expression between each treatment. Figures were plotted in OriginPro 2023b (study version).

## 3. Results

### 3.1. Low-Temperature Treatment

Both *B. dorsalis* and *B. correcta* showed significantly increased survival at low-temperature hardening temperatures, but the optimal induction temperature intervals were different. The survival rate of *B. dorsalis* increased significantly at a hardening temperature of 5 °C, and increasing the temperature to 9 °C further enhanced their cold tolerance, indicating that 9 °C was a key temperature for *B. dorsalis* (Figure 1). The survival rate of *B. dorsalis* was significantly enhanced by its hardening at temperatures > 5 °C, and *B. correcta* showed a high level of cold resistance until 13 °C (Figure 1). In summary, the survival rate of *B. dorsalis* was higher than that of *B. correcta* at 11–22 °C, whereas the survival rate of *B. correcta* was higher than that of *B. dorsalis* at 5–9 °C.

### 3.2. Transcriptome Data Analysis

After sequencing quality control, a total of 176.15 Gb of data were obtained. The percentage of Q30 bases for each sample was greater than 94.70%, indicating reliable sequencing results (Appendix A). A total of 81,695 unigene assemblies were obtained from the sequencing data; unigene N50 was 953 bp, with an average fragment length of 660.05 nucleotides (Appendix A, Appendix A). For all samples, more than 70% of the sequencing data matched the assembly results. According to the results of the principal component analysis (PCA), the different temperatures and biological replicates of *B. dorsalis* and *B. correcta* were separated from each other, indicating that the gene expression of these two species differed more between the different hardening treatments (Appendix A).

Considering the number of differentially expressed genes and proportion of differentially expressed genes in the GO analysis, the first position in the hardening group at 5 °C was for molecular function, and the first position in the hardening group at 9 °C was for cytoplasm, compared to the control group at 25 °C in both *Bactrocera* species (Appendix A). A total of 9776 unigenes were matched to the KEGG database across all unigenes, and the pathways with higher gene enrichment in *B. dorsalis* under the hardening temperature treatment were cutin, suberin, and wax biosynthesis, and the Toll and Imd signaling pathways (Appendix A, Appendix A). The pathways that were most enriched in *B. correcta* were the relaxin signaling, protein digestion, and absorption pathways, whereas thermogenesis showed the highest number of enriched genes (Appendix A). These results suggest different pathway functions in the two *Bactrocera* species under cold conditions (Appendix A). In *B. dorsalis*, 156 genes were differentially expressed at 9 °C compared to the control temperature of 25 °C; of the 156 genes, 122 were up-regulated and 34 were down-regulated, including five up-regulated Hsp genes. In *B. correcta*, 1072 genes were differentially expressed at 5 °C compared to the control temperature of 25 °C; of these 1072 genes, 306 were up-regulated and 766 were down-regulated, including 18 up-regulated and 1 down-regulated Hsp gene (Figure 2 and Appendix A). After screening the differentially expressed genes following rapid cold hardening, *Hsp23*, *Hsp68*, *Hsp70*, and *Hsp27* from the two *Bactrocera* species were selected for gene function studies. These four genes were upregulated at different temperature combinations in both *Bactrocera* species, excluding *Hsp27* in *B. dorsalis*.

### 3.3. Expression Level of Selected Hsp Genes

Expression of *Hsp68*, *Hsp23*, and *Hsp70* increased significantly at the two hardening temperatures compared to that in the 25 °C control group, separately, which was consistent with the transcriptome results. Expression of *Hsp68* was upregulated 15.37-fold and 8.14-fold, *Hsp23* 2.99-fold and 2.84-fold, and Hsp70 6.09-fold and 10.80-fold in *B. dorsalis* at 5 °C and 9 °C, respectively, compared to the control at 25 °C (Figure 3). The expression level of *Hsp70* in the 9 °C hardening group was higher than that in the 5 °C hardening group in particular. In *B. correcta*, expression of *Hsp68* was upregulated 3.29-fold and 1.31-fold, *Hsp23* 7.25-fold and 4.74-fold, and *Hsp70* 1.98-fold and 1.27-fold at 5 °C and 9 °C, respectively, compared to the control at 25 °C (Figure 3). In addition, *Hsp27* expression was consistent with the transcriptome results described above: *Hsp27* expression was significantly increased at 5 °C and 9 °C in *B. correcta* compared to the control, whereas, in *B. dorsalis*, there was no significant difference in *Hsp27* expression at 5 °C and 9 °C.

In terms of absolute expression level, *Hsp23* was the most highly expressed of the four Hsp genes in *B. dorsalis*, whereas *Hsp68* was the most highly expressed of the four Hsp genes in *B. correcta*. The absolute expression levels of *Hsp68*, *Hsp23*, *Hsp70*, and *Hsp27* were much higher in *B. dorsalis* than in *B. correcta* at the different treatment temperatures.

### 3.4. Functions of Hsp Genes

After 96 h of *dsRNA* feeding, RNA interference efficiency increased from 40% to 70% (Figure 4 and Figure 5), and the survival rate of both *B. dorsalis* and *B. correcta* significantly decreased after knockdown of these four *Hsps*.

In *B. dorsalis*, *dsHsp* feeding reduced the expression level of the four Hsp genes compared to that in the *dsGFP* group, although the expression level of these genes was significantly upregulated at the two cold hardening temperatures compared to the control (25 °C). In the *dsHsp23* group, expression of the *Hsp23* gene was significantly elevated under 9 °C and 5 °C hardening, but the survival rate of these two groups had no significance compared to that in the 25 °C group. (Figure 4a,b). The survival rate of the *dsHsp68* group was consistent with *Hsp68* expression level (Figure 4c,d). Larvae’s survival rate in the *dsHsp70*-treated group held at the 5 °C hardening temperature, even though they were fed dsRNA for 96 h, showed a significantly reduced expression of *Hsp70* compared to that in the *dsGFP* group under 5 °C hardening. (Figure 4e,f). In the 9 °C hardening group, the expression level of *Hsp70* was reduced after feeding *dsHsp70*, and the survival rate was also reduced compared to the *dsGFP* group. In the 5 °C hardening group, the expression level of *Hsp27* in the *dsHsp27* group was much higher than that in the *dsGFP* group, but the survival rate was not. Expression of *Hsp27* was not consistent with the survival rate after feeding the larvae *dsHsp27*, although it was abundantly expressed following rapid cold hardening (Figure 4g,h).

In *B. correcta*, feeding *dsRNA* had no effect on the expression of the four Hsp genes compared to that in the control group at 25 °C, and some Hsp genes were more highly expressed than those in the *dsGFP* group after rapid cold hardening. There was a significant increase in the expression of Hsp genes at both hardening temperatures, with a concomitant and significant increase in the survival rate, compared to the 25 °C group. The survival rate of the *dsHsp23* group was consistent with *Hsp23* expression levels. The survival rate was significantly increased at the cold hardening temperatures compared to the control at 25 °C, followed by a significant increase in *Hsp23* expression (Figure 5a,b). In the *dsHsp70*-treated group, feeding *dsHsp* affected Hsp gene expression, with lower expression of *Hsp70* in the *dsHsp70* group than in the *dsGFP* group at the hardening temperature; however, survival rate did not increase as the hardening temperature decreased (Figure 5e,f). For the *dsHsp68* and *dsHsp27* treatment groups, expression levels were restored at both hardening temperatures of 9 °C and 5 °C, respectively, compared to the *dsGFP* group. However, neither group showed higher survival levels than that of the *dsGFP* group (Figure 5c,d,g,h). The expression of these four Hsp genes was highest in both the *dsHsps* and *dsGFP* groups at the optimal hardening temperature for *B. correcta* (5 °C).

## 4. Discussion

Many *Bactrocera* species, including *B. dorsalis* and *B. correcta*, are invasive. Exploring their adaptability under cold hardening may help explain their expansion, which is expected to increase with climate change [13]. This study explored the survival rates of *B. dorsalis* and *B. correcta*, which have expanded their distributions in recent years, under different rapid cold hardening temperatures [32]. The transcriptome data showed differentially expressed genes and important pathways implicated in rapid cold hardening by the two species, and several *Hsps* were identified and their expression levels and gene functions examined.

The low temperature treatment results showed that survival rate trends for the two species differed, as the survival rate of *B. dorsalis* gradually increased above 9 °C and then decreased above 14 °C, whereas the survival rate of *B. correcta* gradually decreased from 5 °C to 15 °C and remained constant, at approximately 75%, from 5 °C to 13 °C. *B. dorsalis* is a widespread species that did not show a survival rate advantage. However, *B. dorsalis* had a wider distribution than that of *B. correcta*, and the northernmost distribution line of *B. dorsalis* was further north than that of *B. correcta* [7,9]. This may be because *B. dorsalis* is more adaptable to changing temperatures in natural environments; their survival ability in natural cold conditions is not only related to their defense capability under one constant hardening temperature, which needs further studies to confirm. Nevertheless, these results reflect differences in the adaptability of the two *Bactrocera* flies under rapid cold-hardening conditions.

Transcriptome sequencing is an important technology that can help researchers identify the key genes involved in biological processes and evaluate gene expression levels in organisms. After identifying the hardening temperature at which the two species showed the highest survival rate, we selected ten larvae per tube after 4 h of cold hardening and 1 h of recovery to test the genes’ expression. The recovery period can ensure the full expression of genes in the larvae. Our results showed that metabolic and other signaling pathways were enriched and associated genes were more abundant in *B. dorsalis*. In contrast, several energy-producing pathways were identified as important to rapid cold hardening in *B. correcta*, indicating that the two fruit flies had different patterns of adaptation under rapid cold hardening conditions. Among the differentially expressed genes identified, genes related to glucose and lipid metabolism showed higher expression levels following cold hardening, which will be verified in our future studies. In this study, after screening, *Hsp68*, *Hsp23*, *Hsp70*, and *Hsp27* were identified as differentially expressed. Expression of the four *Hsp* genes and the survival rate in both *Bactrocera* species increased after cold hardening relative to these measures in the control. In terms of the association between gene expression and larval survival after treatment with extremely low temperatures, after knocking down *Hsp68* and *Hsp70* in *B. dorsalis*, the survival rates were decreased in three different temperature treatment groups (5 °C, 9 °C, and 25 °C), while *Hsp23* and *Hsp70* showed the same patterns of change in *B. correcta*. It is noteworthy that, in *B. dorsalis*, *Hsp70* expression was higher in the 9 °C hardening group than in the 5 °C hardening group, which may account for the significantly higher slope of the 9–10 °C curve (Figure 1). In addition, the absolute expression levels of *Hsp68*, *Hsp23*, *Hsp70*, and *Hsp27* were higher in *B. dorsalis* than in *B. correcta* at different hardening temperatures, which might explain why *B. dorsalis* is a widespread species with a northern range extension. Using RNA interference, *Hsp68* and *Hsp70* were identified as being important for rapid cold hardening in *B. dorsalis*, whereas *Hsp23* and *Hsp70* were identified as being important in *B. correcta*. Previous studies have shown that *Hsp23* is a key gene in the thermal plasticity of *B. dorsalis*, and the *Hsp70* family is important for temperature tolerance in other species, which may be connected to our conclusions [22,23], and we guess the Hsp70 family probably plays an important role in cold hardening in different species. In addition, there were many differentially expressed genes in our transcriptome data that might also play important roles in the low-temperature tolerance of *B. dorsalis* and *B. correcta*. These genes include *Lip3,* which is a low-temperature lipase in mammals. These additional genes warrant further study.

In this study, the cold adaptabilities of two *Bactrocera* species, *B. dorsalis* and *B. correcta*, were assessed and compared. Transcriptome data were obtained from the two species, and their survival rates were assessed under hardening temperatures. Finally, we screened four *Hsp* genes and tested their putative functions in rapid cold-hardening conditions. We confirmed that *Hsp68* and *Hsp70* are involved in rapid cold hardening in *B. dorsalis*, whereas *Hsp23* and *Hsp70* are involved in rapid cold hardening in *B. correcta*. This study offers information about the adaptability of these two fruit fly species to cold temperatures, which may explain their current distribution as well as their potential for geographic expansion.

## Figures and Tables

**Figure 1 insects-15-00090-f001:**
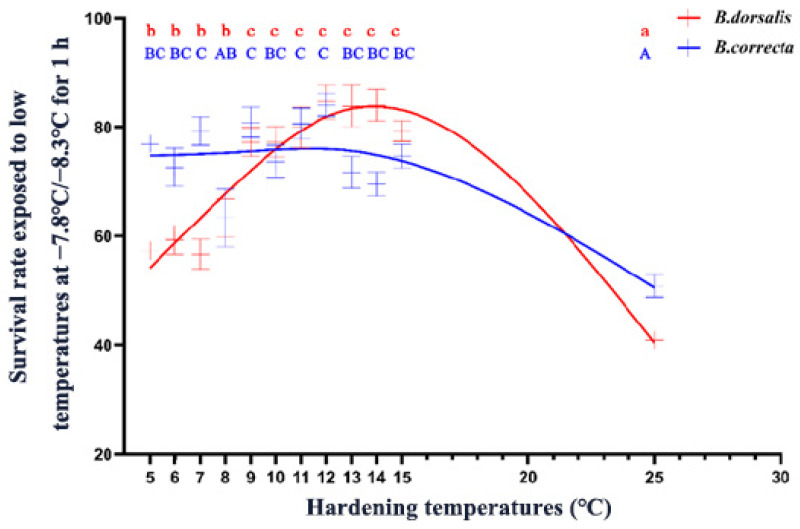
The survival rate of 7-day-old 3rd instar larvae exposed to low temperatures after cold hardening treatments. The red line represents the fitted curve of larval survival rate using an asymmetric model for *Bactrocera dorsalis*. The blue line shows the fitted curve for *Bactrocera correcta* using the same model. Different letters above the bars indicate significant differences at *p* < 0.05, as determined by a Tukey HSD test. The lowercase letters in red refer to *B. dorsalis,* and the capital letters in blue refer to *B. correcta*.

**Figure 2 insects-15-00090-f002:**
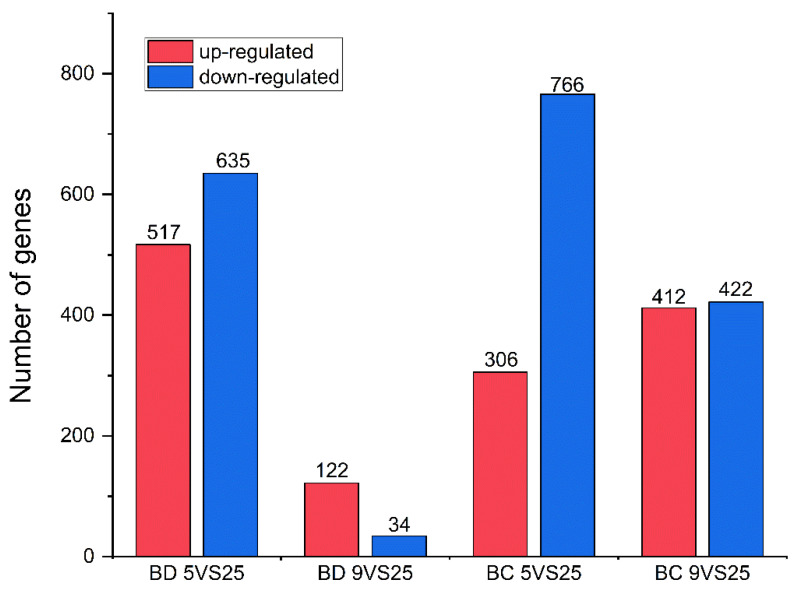
Differently expressed genes (DEGs) in the transcriptome analysis. Up-regulated DEGs are shown in red, while down-regulated DEGs are shown in blue. “BD” indicates *Bactrocera dorsalis,* and “BC” indicates *Bactrocera correcta*.

**Figure 3 insects-15-00090-f003:**
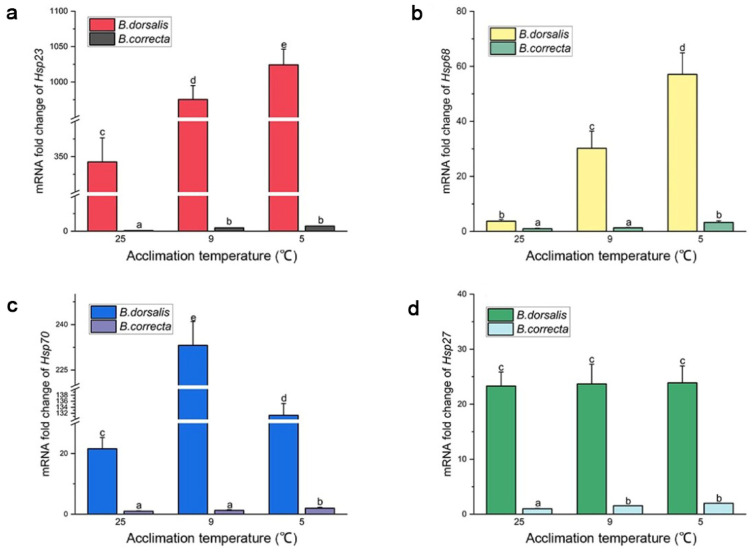
The gene expression level of *Hsps* at two hardening temperatures (5 °C and 9 °C) and the 25 °C control in two *Bactrocera* species. The *Bactrocera correcta Hsps* expression level of the 25 °C group (CK) in each figure was set at 1. (**a**) Expression level of *Hsp23*; (**b**) expression level of *Hsp68*; (**c**) expression level of *Hsp70*; (**d**) expression level of *Hsp27*. Different letters above the bars represent significant differences at *p* < 0.05, as determined by a *t* test. The SEM of each group is indicated above the bars.

**Figure 4 insects-15-00090-f004:**
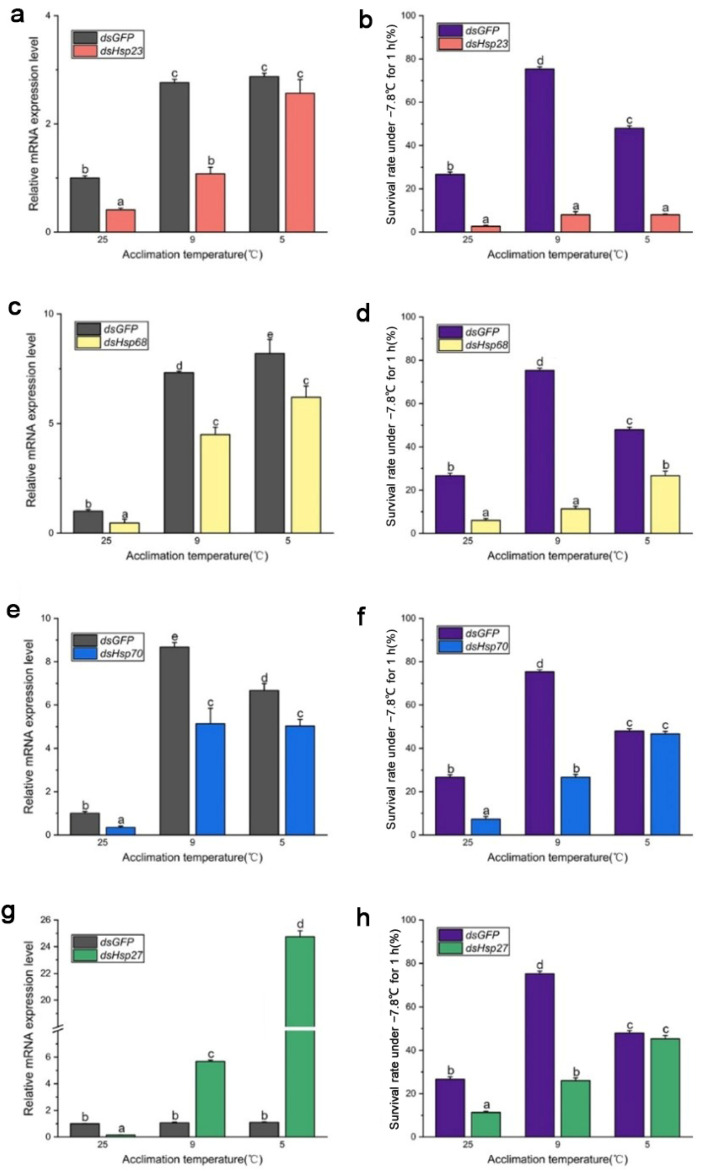
Functional analysis of *Hsp23*, *Hsp68*, *Hsp70*, and *Hsp27* using RNAi in *Bactrocera dorsalis*. (**a**) Expression of *Hsp23* after RNAi in *Bactrocera dorsalis*. (**b**) Survival rate of *Bactrocera dorsalis* under the extreme cold stress of −7.8 °C for 1 h. (**c**) Expression of *Hsp68* after RNAi in *Bactrocera dorsalis*. (**d**) Survival rate of *Bactrocera dorsalis* under the extreme cold stress of −7.8 °C for 1 h. (**e**) Expression of *Hsp70* after RNAi in *Bactrocera dorsalis*. (**f**) Survival rate of *Bactrocera dorsalis* under the extreme cold stress of −7.8 °C for 1 h. (**g**) Expression of *Hsp27* after RNAi in *Bactrocera dorsalis*. (**h**) Survival rate of *Bactrocera dorsalis* under the extreme cold stress of −7.8 °C for 1 h. All the flies of *Bactrocera dorsalis* were fed by *dsHsps* and *dsGFP* for 96 h. Different letters above indicate significant differences at *p* < 0.05, as determined by a *t* test. Each bar shows the SEM of each group.

**Figure 5 insects-15-00090-f005:**
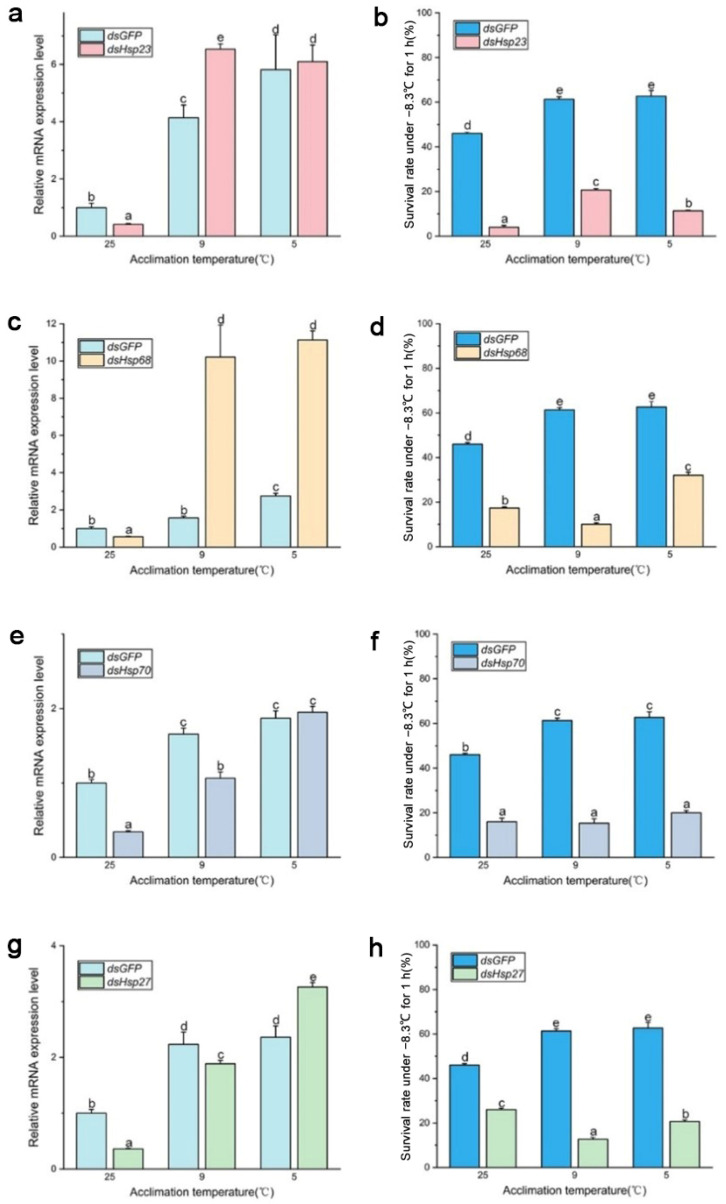
Functional analysis of *Hsp23*, *Hsp68*, *Hsp70*, and *Hsp27* using RNAi in *Bactrocera correcta*. (**a**) Expression of *Hsp23* after RNAi in *Bactrocera correcta*. (**b**) Survival rate of *Bactrocera correcta* under the extreme cold stress of −7.8 °C for 1 h. (**c**) Expression of *Hsp68* after RNAi in *Bactrocera correcta*. (**d**) Survival rate of *Bactrocera correcta* under the extreme cold stress of −7.8 °C for 1 h. (**e**) Expression of *Hsp70* after RNAi in *Bactrocera correcta*. (**f**) Survival rate of *Bactrocera correcta* under the extreme cold stress of −7.8 °C for 1 h. (**g**) Expression of *Hsp27* after RNAi in *Bactrocera correcta*. (**h**) Survival rate of *Bactrocera correcta* under the extreme cold stress of −7.8 °C for 1 h. All the flies of *Bactrocera correcta* were fed by *dsHsps* and *dsGFP* for 96 h. Moreover, *18s* was used as the control gene. Different letters above indicate significant differences at *p* < 0.05, as determined by a *t* test. The SEM of each group is indicated above the bars.

## Data Availability

All data generated and analyzed during the current study are included in this article. The transcriptome raw data may be available from the corresponding author on reasonable request.

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
