# Peer review of "Heat Shock Protein Genes Affect the Rapid Cold Hardening Ability of Two Invasive Tephritids"

_insects, 2024, doi:10.3390/insects15020090_

Round 1

Reviewer 1 Report

Comments and Suggestions for Authors

The manuscript “Heat shock protein genes affect the cold-acclimation ability of two invasive tephritids” submitted by Wang et al. investigated the cold acclimation ability of the two species. Four HSP genes were screened based on cold-treated transcriptome to study their functions. The results explored the adaptability of the two species to cold, and demonstrated the expression and function of four HSPs in response to cold acclimation. In particular, it provides basic theories for the invasion and rapid expansion of the two pest flies. This study is helpful for understanding the mechanisms of the cold adaptation of tephritids. There are several points need to be addressed before acceptance.

Major points:

1.     Although the boundary between acclimation and hardening is not so clear and definite, they are two different plasticity responses. In the manuscript, the authors take 4 h as the acclimation time, is hardening more appropriate? Moreover, line 101 uses cold-hardening experiments, but most in the manuscript is described by cold acclimation, which is contradictory. For details, please refer to the literature “What Can Plasticity Contribute to Insect Responses to Climate Change?” (DOI10.1146/annurev-ento-010715-023859).

2.     The transcriptome sequencing information is not detailed enough, such as the biological replicates of different treatment. In addition, the biological replicates reflected in the PCA results in Figure S2 were inconsistent with those reflected in Table S2.

3.     The primer information given in Table S1 is incomplete and contains errors. Please check carefully and provide details. Gene and primer names are preceded by Latin abbreviations, such as BdoHsp70 and BcoHsp70, which are easier to distinguish in the manuscript.

4.     In order to examine the effects of acclimation or hardening on the thermal tolerance of the fruit flies, there is no problem in the existence of a recovery stage. However, in transcriptome sequencing, after 4 h of treatment at 5℃ and 9℃, recovery was performed for 1 h at 25℃, which included two processes of cold treatment and recovery. This should be mentioned in the discussion part by the authors.

5.     In results 3.3 and 3.4, the description of gene expression quantity, gene silencing efficiency detection and survival rate of the two species under different treatment temperatures was complicated and confusing, and the definition of comparison and control group was not clear. Please readjust the description of the results, such as Lines 221-224, 270-271.

Minor points:

1.     Please provide information about the software version used, such as Primer Premier 6.

2.     Line 183, the objects for upper and lower letters in the Figure 1 legend are reversed.

3.     In the abstract, the 3 in Line 29 should be changed to 4.

4.     Please check lines 306-308 for any inconsistency.

5.     The abbreviation for hour should be the same, h or hr.

6.     The Latin names of species should be italicized, with attention to the use of Latin abbreviations, such as lines 261, 283, and 288.

7.     The significance analysis of the data is not detailed enough and should be supplemented in the Materials and Methods.

Author Response

Dear Reviewer,

Here are my replies about your suggestions and i corrected all of them in my manuscript. Please see the attachment. Many thanks for your notes.

Best wishes,

Yuning

Reviewer 2 Report

Comments and Suggestions for Authors

The manuscript entitled "Heat shock protein genes affect the cold-acclimation ability of two invasive tephritids" investigates the role of heat shock proteins (HSP) in cold tolerance of two species of Tephritidae. Based on transcriptome data, the authors selected several HSP genes and examined their expression in the two Tephritidae species upon exposure to cold. The authors used RNAi to further elucidate the role of these genes in cold acclimation and cold tolerance. The results indicate a contribution of some HSPs to cold tolerance in the two species of Tephritidae.

The present manuscript is interesting and the content is relatively well written, but there are some aspects that need serious attention. Some parts of the methods need to be corrected or clarified. For example, it is not specified what photoperiodic regime the insects were exposed to. The authors also seem to confuse rapid cold hardening with cold acclimation. At the end of the introduction, the authors list objectives, some of which this study clearly cannot meet. At some points in the text, the authors also seem to greatly exaggerate possible uses of the results obtained (e.g., claims that the results will help to predict or prevent the impacts of the species studied on agriculture - I honestly cannot see how they could). The level of English is very inconsistent throughout the text. For example, the Simple summary has poor English, while the Abstract has good English. It almost seems like two people with very different levels of English wrote the text.

Overall, the present manuscript is interesting and suitable for Insects, but some more or less serious issues need to be addressed before acceptance.

Specific comments

Missing description of statistical analysis in Material and methods and elsewhere.

Consider removing the first sentence of the Simple summary.

Figure S10: Add more description. It is not clear which part of the figure belongs to which species.

L11: „…eggs merge into larvae…“ --- Should be EMERGE.

L22-23: “This study provides information to predict and prevent their affect in fruit industry.” --- I'm sorry, but your study provides no such thing.

L48: “variable temperature animals” --- You probably mean “ectothermic”.

L86-89: “The objectives of this work were to help elucidate the adaptation mechanism underlying cold acclimation of the two species, resolve confusion about the differences in geographic range of these two species, and suggest methods to prevent and control the spread of these fruit fly species of agricultural importance.” --- Except for the first point (i.e. the adaptation mechanism), those were clearly not your objectives. You certainly do not suggest any methods to prevent and control the spread of those species. I do not even see how you could when your study is about something else.

L94-95: What was the photoperiodic regime in incubators? It may be important if these species have ability to enter diapause.

L104-106: What you describe seems to be rapid cold hardening, not a cold acclimation (cold acclimation is long-term process that takes days or weeks, not hours; see Teets et al., 2020, Journal of Experimental Biology 223, jeb203448). Expression of HSP genes is one of the known mechanisms that allow rapid cold hardening response in many (perhaps all) insects. Consider calling your “cold acclimation” rapid cold hardening throughout the text.

What do you mean by “extreme low temperature”?

L159-160: Which larvae were exposed to which temperature? The same question for Figure 1.

L293-294: “Exploring their adaptability under cold acclimation may help identify methods to stop their expansion…” --- I really cannot imagine how such knowledge could help to stop their expansion. Can you elaborate?

L319-321: “…which is supported by the results of several previous studies [33,34].”  --- These references refer to other species. How can they support your findings? Different species may show different levels of expression of different HSP genes.

L323-326: I am not sure if I can see these patterns in you results. Maybe I just do not understand what you mean. Please consider rephrasing.

L333-335: “Previous studies…” --- Can you provide a reference/references to those studies?

Comments on the Quality of English Language

The level of English is very inconsistent throughout the text. For example, the Simple summary has poor English, while the Abstract has good English. It almost seems like two people with very different levels of English wrote the text.

Author Response

Dear Reviewer,

Please see the attachment. Thank you again for your comments.

Sincerely,

Yuning

Round 2

Reviewer 2 Report

Comments and Suggestions for Authors

The revised manuscript entitled “Heat shock protein genes affect the rapid cold hardening ability of two invasive tephritids” represents a substantial improvement over the original version. I have identified only a few rather minor issues that I think should be addressed. The language has improved, but I think the text would still benefit from a little editing. I believe the manuscript can be accepted for publication in Insects after a minor revision.

L47: Remove “temperature”. It is just “ectothermic animals”.

L54-57: You seem to mix together the mechanisms of rapid cold hardening with those of diapause and seasonal acclimatization (cold acclimation). Diapause (dormancy) and cold acclimation are adaptations for winter. Rapid cold hardening is rather an adaptation to sudden short-lasting drops in temperature outside the winter season. Even tropical insects that never experience winter are capable of rapid cold hardening. You can keep the sentence largely unchanged, but I suggest removing any mentions about “changes in behavior”, “entering a dormant state” and “reducing water content” as these are not related to rapid cold hardening.

Comments on the Quality of English Language

The language has improved, but I think the text would still benefit from a little editing.

Author Response

Dear Reviewer,

Please see the attachment. Thank you again for your comments and suggestions.

Best wishes, 

Yuning
